# Myeloid PGGT1B Deficiency Promotes Psoriasiform Dermatitis by Promoting the Secretion of Inflammatory Factors

**DOI:** 10.3390/ijms26104901

**Published:** 2025-05-20

**Authors:** Shanshan Yu, Fangyuan Long, Xuecui Wei, Heng Gu, Zhimin Hao

**Affiliations:** 1Institute of Dermatology, Chinese Academy of Medical Sciences & Peking Union Medical College, Nanjing 210042, China; yss0818@njmu.edu.cn (S.Y.); lfy950419@foxmail.com (F.L.); 2School of Public Health, Nanjing Medical University, Nanjing 211166, China; estangs@njmu.edu.cn

**Keywords:** PGGT1B, psoriasis, NF-κB signal, inflammatory

## Abstract

Psoriasis pathogenesis involves dysregulated immune responses, yet the role of protein prenylation (particularly PGGT1B-mediated geranylgeranylation) in macrophage-driven inflammation remains poorly understood. This study aims to explore the role and molecular mechanism of protein geranylgeranyltransferase type I subunit beta (PGGT1B) in the development of psoriasis. Myeloid cell-specific PGGT1B gene knockout mice were generated, and a mouse psoriasis model was established with imiquimod to study the role and mechanism of PGGT1B gene downregulation-induced macrophage activation in the pathogenesis of psoriasis. Bone marrow-derived macrophages (BMDMs) from wild-type and PGGT1B knockout mice were cultured and stimulated with resiquimod (R848) to simulate the immune microenvironment of psoriasis. In addition, the differentially expressed genes induced by PGGT1B knockout were analyzed using RNA-seq, and bioinformatics analysis was carried out to study the possible biological process of PGGT1B regulation. Finally, PMA-THP-1 was co-cultured with HaCaT cells to study the effect of PGGT1B deletion in macrophages on the proliferation and differentiation of keratinocytes. Bone marrow PGGT1B deficiency aggravated the psoriasis-like lesions induced by imiquimod in mice. In BMDMs with PGGT1B deficiency, the NF-κB signaling pathway was over-activated by R848, and the expressions of proinflammatory cytokines IL-1β, IL-6, and TNF-α were significantly increased. Activation of cell division cycle 42 (CDC42) may mediate the activation of the NF-κB pathway in PGGT1B-deficient BMDMs. PGGT1B deletion can promote the proliferation and inhibit the differentiation of HaCaT cells. Reduced PGGT1B levels can increase the expression of CDC42, which further activates NLRP3 inflammation in macrophages through NF-κB signaling, further aggravating the inflammatory state of psoriasis. Psoriasis-like lesions induced by IMQ are aggravated when PGGT1B expression is reduced in mouse bone marrow cells. A possible mechanism for this is that PGGT1B-deficient macrophages migrate to the epidermis more easily during psoriasis, which leads to the activation of Cdc42, NF-κB signaling, and NLRP3 inflammatory corpuscles.

## 1. Introduction

Psoriasis is a chronic, recurrent, and inflammatory skin disease that affects about 2–3% of the global population, making it one of the most common chronic skin diseases in the world [1]. The disease is characterized by erythema, scales, and itching, which seriously affects the quality of life of patients, including skin symptoms, mental health, and social interaction. According to its clinical features, psoriasis can be classified as the common, arthropathy, pustule, or erythroderma type. Psoriasis vulgaris is the most common, accounting for more than 99% of cases. Its main pathological features include epidermal hyperproliferation, inflammatory cell infiltration, and dermal vasodilation. These pathological changes are mainly caused by immune abnormalities induced by genetic and environmental factors, involving keratinocytes and various immune cells, such as T cells, plasma-like dendritic cells, myeloid dendritic cells, neutrophils, and macrophages, which together form an inflammatory circuit and promote the onset and development of psoriasis [2,3,4]. These cells secrete a large number of proinflammatory cytokines, such as tumor necrosis factor α (TNF-α), interferon γ (IFN-γ), and interleukin-17 (IL-17), which leads to abnormal proliferation and persistent inflammatory reaction of keratinocytes [2,3]. Although remarkable progress has been made in the treatment of psoriasis in recent years, the understanding of its pathogenesis is still limited. At present, the treatment methods for psoriasis mainly include local medication, phototherapy, and systematic treatment. Topical medication, such as glucocorticoids and vitamin D analogs, is the first choice of treatment for mild psoriasis. Phototherapy, such as ultraviolet B (UVB) and narrow-band ultraviolet B (NB-UVB), is effective for moderate and severe psoriasis. Systematic therapy includes traditional drugs, such as methotrexate, cyclosporine, and avia, as well as biological agents, such as anti-TNF-α antibodies and IL-17 inhibitors [5,6]. However, these treatments have some limitations. Therefore, it is of great significance to study the pathogenesis of psoriasis and find new therapeutic targets.

Protein geranylgeranylation is a kind of lipoprotein modification that plays a key role in regulating the membrane localization and activation of coenzyme A reductase. PGGT1B, also known as BGGI or GGT, is an enzyme involved in the acylation of protein amino acid geranyl geranium. Under normal circumstances, the acylation of protein geranyl geranium helps to maintain the normal function of immune cells. However, when PGGT1B is defective, it may affect this modification process, resulting in abnormal immune cell function [7]. Studies have shown that myeloid PGGT1B deficiency results in the abnormal function of immune cells [8]. For example, the deletion of PGGT1B in mouse intestinal epithelial cells leads to cytoskeleton rearrangement and tight junction redistribution, thus preventing cell detachment, further attracting neutrophils and CD4+ T cells, and finally inducing chronic intestinal inflammation [9]. In addition, the defect of PGGT1B in T cells damages the function of RhoA, increases the expression of α4β7, and leads to an increase in CD4+ T cells in the colon, which leads to the over-expression of Th17-related inflammatory cytokines and mediates the development of colitis [10]. Knocking out PGGT1B in vascular smooth muscle cells can reduce Rac1 activity and ERK1/2 and JNK phosphorylation levels; inhibit vascular smooth muscle cell proliferation, NADPH oxidase activity, and oxidative damage; and effectively inhibit the accelerated atherosclerosis process of diabetes in mice [11]. On the other hand, the conditional deletion of PGGT1B in mouse macrophages increases Rac1 activity; the phosphorylation level of p38, IKK, Src, and STAT3; and the expression of proinflammatory cytokines IL-6, IL-1, and TNF-α after LPS stimulation, which leads to the development of chronic erosive rheumatoid arthritis in mice [12]. These inflammatory factors also play an important role in the pathogenesis of psoriasis. They can stimulate the proliferation and inflammatory reaction of keratinocytes, thus aggravating the condition of psoriasis [13].

The purpose of this study is to explore the role and molecular mechanism of PGGT1B in the development of psoriasis. We established a mouse model of psoriasis induced by imiquimod by constructing myeloid cell-specific PGGT1B gene knockout mice and combined it with RNA-seq analysis to study the role of macrophage activation induced by PGGT1B gene downregulation in the pathogenesis of psoriasis. The results showed that PGGT1B deficiency in myeloid cells aggravates imiquimod-induced psoriasis-like lesions, accompanied by excessive secretion of inflammatory factors. The results of this study help reveal the pathophysiological process of psoriasis and provide a new target for treatment.

## 2. Results

### 2.1. Myeloid PGGT1B Deficiency Exacerbates IMQ-Induced Psoriatic Rash in Mice

In order to study the relationship between PGGT1B deficiency and rash in psoriasis, we used myeloid cell-specific PGGT1B knockout mice as the conditional knockout (cko) group, and wild *Pggt1b*^fl/fl^Lyz2^wt/wt^ mice as the control group (wt group). The acute psoriasis model was established by applying IMQ to the back of mice for 5 days after depilation, and the control group was given the same dose of Vaseline (VAS) ointment. When the skin injury reached its highest severity, samples were photographed and examined using histopathology, IHC, mRNA analysis, and skin homogenate (Figure 1A). The results show that compared to the VAS group, erythema and a small amount of scales and wrinkles appeared around the coated area on the wt mice one day after modeling. When the drug was used continuously for 4–5 days, erythema and scale thickening were the most obvious. On the other hand, compared to the wt mice, the cko mice displayed an earlier appearance of rash, more obvious erythema, more scales, and a thicker rash appearance (Figure 1B). At the same time, the psoriasis area and severity index (PASI) score (evaluating erythema, scales, and hypertrophy) of the cko mice was significantly higher (Figure 1C, *p* < 0.05). Compared to the wt group, after IMQ treatment, the cko group had more obvious hyperkeratosis, significantly increased epidermal thickness, enlarged epidermal processes, and proliferated inflammatory infiltrating cells in the dermal papilla (Figure 1D, *p* < 0.05). After the IMQ model was established, compared to the wt group, the rate of Ki67-positive cells in the cko group increased (Figure 1E, *p* < 0.05). In addition, loricrin in the cuticle of the two genotypes of mice decreased, and the decreasing trend in the cko group was more significant than that in the wt group (Figure 1F, *p* < 0.05), which indicates that in the IMQ-induced psoriasis model, myeloid PGGT1B deficiency promoted the proliferation of keratinocytes and inhibited their differentiation.

### 2.2. Myeloid PGGT1B Deficiency Aggravated Psoriasis-like Inflammation Induced by IMQ

In order to explore the relationship between PGGT1B deficiency and inflammation in psoriasis, we measured the mRNA and cytokine expressions of inflammatory factors IL-1β, IL-6, TNF-α, IL-17A, and IL-10 in the most serious stage of skin injury in mice using Luminex and qRT-PCR. The results show that the levels of IL-1β, IL-6, TNF-α, and IL-17A in the IMQ group were higher than those in the VAS group, while the level of IL-10 in the IMQ group was lower than that in the VAS group. In addition, in the psoriasis group induced by IMQ, the levels of IL-1β, IL-6, TNF-α, and IL-17A were higher in the cko group than those in the wt group, especially IL-1β and IL-6 (Figure 2A,B,D–G,I,J, *p* < 0.05). The mRNA level of IL-10 decreased, but there was no significant difference (Figure 2C,H, *p* > 0.05). The main sign of NF-κB activation is the transfer of NF-κB from the cytoplasm to the nucleus, where it combines with a specific DNA sequence to start the transcription of related genes. In this study, the IHC staining results of skin sections of the four groups of mice show that NF-κB staining was positive in the cytoplasm of the two groups of VAS mice. Nuclear translocation was observed in epidermal cells, infiltrating inflammatory cells, and some epidermal cells of the IMQ mice in both groups. The number of NF-κB nuclear translocations in the cko group was significantly higher than that in the wt group (Figure 2K, *p* < 0.05), indicating that the lack of myeloid PGGT1B aggravated the psoriasis-like inflammation induced by IMQ. In order to clarify the role of myeloid cells in this process, IHC staining of Ly6G (neutrophil marker) and F4/80 (macrophage marker) was performed on the skin lesions of each group of mice. The results show that there was no statistical difference in the number of Ly6G-positive stained cells between the two genotypes (Figure 2L, *p* > 0.05). After 5 days of IMQ coating, the positive rate of F4/80 in the mouse dermis and dermal papilla increased significantly, and a small number of positively stained cells was also observed in the epidermis (Figure 2M, *p* < 0.05). These results show that PGGT1B regulated the activation of macrophages, affecting the intensity of the inflammatory response in psoriasis.

### 2.3. PGGT1B Deficiency Promotes the Secretion of Proinflammatory Factors and Inhibits the Secretion of Anti-Inflammatory Factors in BMDMs

To investigate the effects of myeloid-specific PGGT1B knockout on the development of psoriasis, we stimulated BMDMs with R848 to mimic the immune microenvironment of psoriatic macrophages in mice. Then, transcriptome sequencing was employed to identify the alterations in biological processes induced by PGGT1B knockout (Figure 3A). Gene analyses of four binary comparison groups (wtcon, wtR848, ckocon, and ckoR848) were performed. To identify the DEGs resulting from PGGT1B deficiency in response to R848 stimulation, we compared three sets of differentially expressed genes: wtR848 vs. wtcon, ckoR848 vs. ckocon, and ckoR848 vs. wtR848. The results reveal 226 DEGs (Figure 3B–D). Next, the differentially expressed gene–protein interaction network was analyzed. For the species within the database, the interaction relationships of the target genome were directly extracted to construct the network (Figure 3E). The MCODE plug-in was used to identify subnet clusters. All the genes with the highest score in the network were considered the central gene (cluster1), and the results are listed in Table 1 (Figure 3F). Central genes promote the biological process of cytokine binding to cytokine receptors by affecting the molecular function of mediated signaling pathways. In addition, the concentrations of proinflammatory factors IL-1β, IL-6, TNF-α, and IL-17A in the cko group were significantly higher than those in the wild-type group 12 h after R848 stimulation of bone marrow stromal cells. The IL-10 in the ckoR848 group was significantly lower than that in the wtR848 group (Figure 3G–K, *p* < 0.05). The mRNA expression of the hub gene was verified using qPCR (Figure 3L–P, Appendix A). This suggests that PGGT1B deficiency promotes the secretion of proinflammatory factors and inhibits the secretion of anti-inflammatory factors in BMDMs.

### 2.4. PGGT1B Defects Promoted NLRP3 Inflammasome Activation and IL-1β Secretion Through NF-κB Signaling Pathway

In order to explore the effect of bone marrow-specific PGGT1B knockout on the NF-κB signaling pathway, we found the top 10 pathways related to cytokine secretion through KEGG enrichment analysis of hub genes. These pathways include cytokine–cytokine receptor interaction, the chemokine signaling pathway, the toll-like receptor signaling pathway, the NF-κB signaling pathway, and the TNF signaling pathway (Figure 4A). In addition, a protein blot was used to detect the expression of p65 in BMDM whole cell lysate at different time points after R848 stimulation. The results show that after 5 min, the expression of p-p65 in both groups began to increase and, after 15 min, the expression of PGGT1B-deficient BMDM was higher than that in the wild-type group (Figure 4B, *p* < 0.05). At the same time, after 24 h of R848 stimulation, the expressions of NLRP3, ASC, caspase-1, p-p65, IL-1β, and cleaved-caspase-1 in the whole cell lysate and the activation of NLRP3 inflammatory vesicles in the supernatant were analyzed. There was no significant difference in ASC expression between the two groups. The expression of NLRP3 increased in both groups, especially in PGGT1B-deficient cells. Compared with the wild-type BMDM, the expressions of cleaved-caspase-1 and IL-1β were significantly higher in the PGGT1B-deficient BMDM cells and culture supernatant (Figure 4C). Next, we pretreated BMDMs with or without the NF-κB pathway inhibitor JSH-23. In addition, we used MCC950, an inhibitor of NLRP3, to further clarify whether PGGT1B affects the expression of IL-β through the activation of NLRP3 inflammatory corpuscles. The results show that compared to the R848 group, the expressions of p-p65, NLRP3, and IL-1β in the supernatant of the JSH+R848 group were significantly decreased. Compared to the R848 group, cleaved-caspase-1 in the MCC950+R848 group decreased significantly in the supernatant, while the intracellular expressions of IL-1β, p-p65, NLRP3, and cleaved-caspase-1 decreased slightly (Figure 4D). These results indicate that JSH-23 directly inhibits the production and secretion of IL-1β in cells, while MCC950 inhibits the secretion of IL-1β by inhibiting the production of cleaved-caspase-1 and reducing the maturation of IL-1β. This suggests that PGGT1B deficiency promotes the activation of inflammatory corpuscles and the secretion of IL-1β in NLRP3 by activating the NF-κB signaling pathway.

### 2.5. PGGT1B-Deficient Macrophages Promoted the Proliferation of HaCaT Cells in Contact Culture

In order to study the effect of PGGT1B-deficient macrophages on the proliferation and differentiation of keratinocytes, we established a co-culture system of macrophages and keratinocytes in vitro. EDU analysis, Ki-67 immunofluorescence analysis, and RT-QCPR analysis were used to observe the effect of PGGT1B on the proliferation and differentiation of HaCaT (Figure 5A). We generated PGGT1B knockout THP-1 cell lines (Figure 4B). We used R848 at concentrations of 0, 0.5, 1, and 2 μg/mL to stimulate HaCaT cells. After 48 h, the cell proliferation rate was determined using the EDU test. The results show that 0.5 μg/mL R848 had no significant effect on the growth of HaCaT cells, while 1 μg/mL R848 led to a significant increase in the proliferation rate of HaCaT cells. Therefore, the concentration of R848 was selected as 0.5 μg/mL in the subsequent experiments (see Appendix A). The expressions of inflammatory factors after R848 stimulation were detected through qRT-PCR, and their levels were consistent with those in the PGGT1B-deficient BMDMs (see Appendix A). In addition, we found that when these two types of cells were in direct contact, the cell proliferation and K17 mRNA level of HaCat cells co-cultured with PGGT1B knockout macrophages were significantly higher than those of wild-type macrophages (Figure 5G–H). At the same time, the expression of K14 mRNA in HaCaT cells co-cultured with wild-type macrophages increased slightly, but there was no statistically significant difference. However, with PGGT1B knockout, the expression of K14 mRNA in HaCaT cells increased significantly. This suggests that macrophages with PGGT1B gene knockout can promote the proliferation of HaCaT, which may be achieved through intercellular anchoring connection and communication connection.

### 2.6. Cdc42 Activation Mediated the Secretion of Proinflammatory Factors in PGGT1B-Deficient BMDMs

In order to study how the acylation of geranium protein regulates NF-κB activity, we used small interfering RNA (siRNA) to inhibit the expression of Cdc42, Kras, or Rhoa in BMDMs with PGGT1B deficiency (Figure 6A), followed by stimulation with R848. The results show that the phosphorylation level of p65 decreased significantly in BMDMs with PGGT1B deficiency after downregulating Cdc42. The inhibition of Kras and RhoA had no effect on p65 phosphorylation (Figure 6B). This suggests that Cdc42 is more easily activated in PGGT1B-deficient cells, which leads to the overactivation of the NF-κB signaling pathway stimulated by R848. This, in turn, increases the production of inflammatory cytokines in macrophages. Figure 6C presents a diagram of this mechanism.

## 3. Discussion

In this study, it was found that PGGT1B deficiency in myeloid cells aggravates imiquimod-induced psoriasis-like lesions, accompanied by excessive secretion of inflammatory factors. These results suggest that PGGT1B may play a role in inhibiting the inflammatory response in myeloid cells, and its absence leads to an uncontrolled inflammatory response, thus aggravating the condition of psoriasis.

Previous studies have found that the expression of PGGT1B in peripheral blood mononuclear cells of patients with psoriasis decreased significantly, suggesting that this enzyme may be involved in the pathogenesis of psoriasis [7]. However, the specific method and mechanism of action are still unclear. In this study, bone marrow cell-specific PGGT1B knockout mice were constructed using gene knockout technology, which further confirmed the key role of PGGT1B in psoriasis. By comparing the performance of wild-type and PGGT1B knockout mice in the psoriasis model, we found that the lack of PGGT1B in bone marrow significantly aggravated the psoriasis-like lesions induced by imiquimod in mice. This result preliminarily confirmed our hypothesis that PGGT1B plays a protective role in the pathogenesis of psoriasis.

The level of inflammatory factors is closely related to the onset and severity of psoriasis. Among them, IL-17 mainly comes from proinflammatory cells in diseased skin, such as mast cells, T cells, and innate lymphoid cells. These cells can stimulate the production of various antimicrobial peptides, chemokines, and factors that promote inflammation and cell proliferation, thus aggravating the inflammatory reaction in tissues [14,15]. In addition, IL-17 can also promote the production of IL-6, which can not only activate and attract more inflammatory cells to the damaged area but also lead to abnormal skin desquamation and hyperkeratosis, which further promotes the formation and development of psoriasis plaques [16,17]. TNF-α also plays an important role in psoriasis. It can induce more proinflammatory cytokines by activating macrophages and T cells. The inhibition of TNF-α has been proven to have a significant effect on relieving psoriasis symptoms [18,19]. In this study, in order to explore the relationship between PGGT1B deficiency and inflammation in psoriasis, we also detected the mRNA and cytokine expressions of inflammatory factors IL-1β, IL-6, TNF-α, IL-17A, and IL-10 in the most serious stage of skin injury in mice. Our experimental results show that the expressions of proinflammatory cytokines IL-1B, IL-6, and TNF-α were significantly increased. The overexpression of these inflammatory factors is one of the important characteristics in the pathogenesis of psoriasis, which is consistent with previous research results showing that the loss of PGGT1B may lead to an imbalance of the immune system, thus aggravating the inflammatory response. Then, BMDMs were stimulated by R848 to simulate the immune microenvironment of psoriasis. R848 is a synthetic TLR7/8 agonist, similar to IMQ, which can activate the TLR7/8 signaling pathway and induce immune cells to produce an inflammatory reaction. During in vitro experiments, R848 is often used as a stimulator of BMDMS (bone marrow-derived macrophages) to simulate the immune activation state in the psoriasis microenvironment. Compared with IMQ, R848 is more convenient to operate and has its role in vitro experiments. We found that the NF-κB signaling pathway was overactivated in BMDMs with PGGT1B deficiency. These results indicate that PGGT1B may affect the development of psoriasis by regulating the NF-κB signaling pathway.

NF-κB is a widely studied signal transduction pathway that plays a central role in regulating the inflammatory response, immune response, cell proliferation, and apoptosis. P65 is a member of the NF-κB family, which usually forms a heterodimer with p50. In unstimulated cells, this dimer is bound to the cytoplasm by the inhibitory protein IκB. When cells are stimulated by external stimuli (such as TNF-α, IL-1β, LPS, etc.), IκB kinase (IKK) is activated, which leads to phosphorylation and the subsequent degradation of IκB protein, releasing p65/p50 dimer. The released p65/p50 complex can enter the nucleus and start the transcription of downstream genes [20,21]. In order to further explore the possible biological process of PGGT1B regulation, we conducted RNA-seg analysis and found differentially expressed genes induced by PGGT1B knockout. Through bioinformatics analysis, we found that these differentially expressed genes are closely related to biological processes, such as the NF-KB signaling pathway, cell proliferation, and differentiation. At the same time, we detected the expression of p65 in BMDM whole cell lysates at different time points after R848 stimulation. The results show that 5 min after R848 stimulation, p-p65 in both groups began to increase, indicating that the NF-κB signaling pathway was activated rapidly. However, after 15 min, the expression of p-p65 in PGGT1B-deficient BMDMs was significantly higher than that in the wild-type group. This result indicates that the loss of PGGT1B may aggravate the inflammatory reaction by promoting or prolonging the activation of the NF-κB signaling pathway. After 24 h of R848 stimulation, the expressions of NLRP3, ASC, caspase-1, p-p65, IL-1β, and cleaved-caspase-1 in whole-cell lysates and the activation of NLRP3 inflammatory vesicles in the supernatant were further analyzed. These results indicate that the loss of PGGT1B not only affects the NF-κB signaling pathway but also promotes the activation of NLRP3 inflammatory corpuscles. The NLRP3 inflammatory corpuscle is an important inflammatory regulator that can promote the maturation and release of proinflammatory cytokines, such as IL-1β, after activation, thus aggravating the inflammatory state of psoriasis. These results suggest that PGGT1B plays an important role in maintaining macrophage homeostasis and preventing excessive inflammatory reaction.

In the process of exploring the regulatory mechanism of PGGT1B, we also found that the activation of cell division cycle 42 (Cdc42) may mediate the activation of the NF-κB pathway in PGGT1B-deficient BMDMs. CDC42 is an important cytoskeleton protein involved in regulating biological processes, such as cell morphology, polarity, and migration [22]. Previous studies have found that increased serum Cdc42 levels in psoriasis patients reflect disease severity [23]. In this study, in BMDMs with PGGT1B deficiency, the phosphorylation level of p65 decreased significantly after downregulating Cdc42. This suggests that the activation of Cdc42 may be one of the key links that PGGT1B deletion leads to overactivation of the NF-κB signaling pathway and an increase in inflammatory factor expression. However, at present, it is not clear whether PGGT1B’s influence on Cdc42 is direct or indirect. Cdc42 belongs to the small G protein family, and many members of this family (such as Ras and Rac) are known to undergo isopentenylation modification. This modification usually occurs on the C-terminal cysteine residue. By adding isoprene groups (such as farnesyl or geranyl geranyl), the combination of protein and cell membrane is promoted, thus affecting its localization and function. PGGT1B may indirectly regulate the activity of Cdc42 by influencing geranyl geranyl acylation of other small G proteins. Importantly, it is crucial to acknowledge that direct evidence of Cdc42 isopentenylation remains lacking, and further investigation is required to determine whether Cdc42 itself is a direct substrate of PGGT1B. In addition, PGGT1B may indirectly participate in the pathogenesis of psoriasis by affecting the proliferation and differentiation of keratinocytes. Our results show that PGGT1B deletion can promote the proliferation and inhibit the differentiation of HaCaT cells, which may be closely related to the pathological characteristics of excessive proliferation and abnormal differentiation of keratinocytes in psoriasis lesions. This discovery further emphasizes the important role of PGGT1B in maintaining skin homeostasis.

The advantage of this study is that we use a knockout mouse model, which can specifically study the role of PGGT1B in myeloid cells and avoid other interference factors that may be caused by systemic knockout. At the same time, the mouse model of psoriasis induced by imiquimod can simulate the pathological process of human psoriasis well and provide a powerful tool for studying the pathogenesis of psoriasis. In addition, we combined RNA-seq analysis to fully understand the effect of PGGT1B deficiency on the gene expression of myeloid cells, which provided clues for further study of its mechanism. However, this study also has some limitations. First of all, this study is mainly based on animal models, and whether the results are applicable to humans needs further verification. Although the mouse model of psoriasis induced by imiquimod can simulate some characteristics of human psoriasis, there are still differences between it and human psoriasis in etiology and pathology. Therefore, future research needs to include human psoriasis skin or PBMC datasets and verify the expression level and function of PGGT1B in patients with psoriasis in clinical samples. Secondly, this study mainly focuses on the role of PGGT1B in myeloid cells while ignoring the possible role of other cell types (such as keratinocytes, T cells, etc.). Psoriasis is a complex disease, involving the interaction of many cell types. Therefore, future research needs to explore the role of PGGT1B in different cell types and its overall role in the pathogenesis of psoriasis. Furthermore, a key limitation is the lack of direct evidence demonstrating Cdc42 as a direct target of PGGT1B. Future studies should focus on elucidating whether Cdc42 undergoes geranylgeranylation by PGGT1B and the functional consequences of this modification. In addition, the specific molecular mechanism of PGGT1B regulating the inflammatory response of myeloid cells has not been explored in this study. Future research can further explore how PGGT1B affects the signal pathway, transcription factor activity, and inflammatory factor secretion of myeloid cells, thus providing a theoretical basis for developing therapeutic strategies for PGGT1B.

## 4. Materials and Methods

### 4.1. Animals

#### 4.1.1. Animal Origin

*Pggt1b*^em1Cflox^/Gpt mice were obtained from GemPharmatech Co., Ltd. (Nanjing, China) *Pggt1b*^fl/fl^Lyz2^iCre/wt^ and littermate control Pggt1b^fl/fl^Lyz2^wt/wt^ mice were generated by first crossing the *Pggt1b*^em1Cflox^/Gpt with B6/JGpt-Lyz2^em1Cin(iCre)^/Gpt mice and then intercrossing the resultant *Pggt1b*^fl/fl^Lyz2-Cre mice. Littermate controls were used in the experiments. The mouse strains were maintained in specific pathogen-free conditions in the Experimental Animal Center, Institute of Dermatology, Chinese Academy of Medical Sciences, and Peking Union Medical College, and the animal study was approved by the Ethics Committee of Dermatology Hospital, Chinese Academy of Medical Sciences (22-DW-007).

#### 4.1.2. Construction of the Psoriasis Mouse Model

This study used a recognized mouse model of IMQ-induced psoriasis [24]. Briefly, healthy C57/B6 mice aged 6–8 weeks were selected for back shaving 1 day before the experiment. The shaving area was 2.5 cm × 3 cm, and hair removal cream was used to remove hair. IMQ was applied to the shaved back skin of mice at a dose of 62.5 mg per day for 5 consecutive days to establish a psoriasis mouse model, while the control group was given the same dose of VAS ointment. The PASI score and pathological changes were evaluated to confirm the success of the model and to evaluate the severity of psoriasis.

### 4.2. Cells

#### 4.2.1. Cell Culture

The immortalized human keratinocyte cell line HaCaT was obtained from the China Center for Type Culture Collection (Wuhan, China) and maintained in Dulbecco’s modified Eagle’s medium (DMEM, Welgene, Daegu, Republic of Korea) supplemented with 10% fetal bovine serum (FBS, Gibco, Carlsbad, CA, USA) and 1% antibiotics (100 U/mL of penicillin, 100 μg/mL of streptomycin). THP-1 cells, the human monocytic cell line, were purchased from the American Type Culture Collection (ATCC, Manassas, VA, USA) and maintained in RPMI-1640 (Welgene, Gyeongsan-si, Republic of Korea) supplemented with 10% FBS and 1% antibiotics. THP-1 cells were induced to differentiate into macrophages using 50 ng/mL of phorbol 12-myristate 13-acetate (PMA, Sigma-Aldrich, St. Louis, MO, USA) in basal medium for 24 h.

#### 4.2.2. Co-Culture of PMA-THP-1 with HaCaT

##### Direct Contact Co-Culture of PMA-THP-1 with HaCaT

THP-1 cells cultured in 24-well plates at a density of 2.5 × 10^5^ cell/mL or 0.5 mL/well were induced with PMA (50 ng/mL) for 48 h, and then the medium was replaced with PMA-free culture medium. Logarithmically grown HaCaT cells were added to the PMA-THP-1 cells at a concentration of 5 × 10^4^ cells/mL or 0.5 mL/well. The next day, the medium in each well was replaced with a complete culture medium containing R848 (0.5 µg/mL) for 48 h.

##### Indirect Contact Co-Culture of PMA-THP-1 with HaCaT

THP-1 cells were cultured in 24-well Transwell plates at a density of 5 × 10^5^ cell/mL, with 0.25 mL/well, and induced with PMA (50 ng/mL) for 48 h. Then, the medium was replaced with PMA-free complete culture medium, and logarithmically grown HaCaT cells were added to the wells below the Transwell inserts at a concentration of 5 × 10^4^ cells/mL, with 0.5 mL/well. The next day, the medium in both the upper and lower chambers of each well was replaced with a complete culture medium containing R848 (0.5 µg/mL) for 48 h.

#### 4.2.3. BMDM Culture and Stimulation

Bone marrow cells were harvested from 6–8-week-old wild-type and conditional knockout mice by flushing the femurs and tibias with PBS, and the cells were cultured in DMEM (C11995500BT; Gibco, Waltham, MA, USA) with 10% FBS (10270106, Gibco) and 20% L929 supernatants for 7 days. BMDMs were stimulated with 1 μg/mL of resiquimod (R848) (S28463; MedChemExpress, Monmouth Junction, NJ, USA) or DMSO (D2650; Sigma-Aldrich, St. Louis, MO, USA). The experiment comprised the following four groups: wild-type BMDMs stimulated with DMSO (wtcon), wild-type BMDMs stimulated with R848 (wtR848), Pggt1b-deficient BMDMs stimulated with DMSO (ckocon), and Pggt1b-deficient BMDMs stimulated with R848 (ckoR848). After 5 h, the cells were collected for quantitative real-time analysis. BMDMs were collected at 0, 5, 15, 30, 60, and 120 min after stimulation for signaling pathway validation using Western blot analysis.

### 4.3. Quantitative Reverse Transcription–Polymerase Chain Reaction (qRT-PCR)

The details of qRT-PCR are described in the Appendix A and Methods Section. The gene-specific primers used in this study are listed in Table 2 and Appendix A.

### 4.4. Immunohistochemistry, HE Staining Experiment Procedure, Luminex Detection Technology, Bioinformatics Analysis, and Western Blot Analysis

The details of immunohistochemistry, HE staining experiment procedure, Luminex detection technology, bioinformatics analysis, and Western blot analysis are described in the Appendix A and Methods Section.

### 4.5. Statistical Analysis

Statistical analysis was performed using SPSS 22.0 software and Prism7.0 software. The data are presented as the mean and SD of at least three independent experiments. A two-sided *t*-test was used for two groups, and one-way ANOVA was used to compare three groups. Statistical significance was observed at *p* < 0.05.

## 5. Conclusions

In summary, this study explored the role and molecular mechanism of PGGT1B in the pathogenesis of psoriasis by establishing myeloid cell-specific PGGT1B gene knockout mice and imiquimod-induced psoriasis mice models. It was found that the deletion of PGGT1B significantly aggravated the psoriasis-like lesions in mice, which was mainly due to the fact that macrophages with PGGT1B deletion were more likely to migrate to the epidermis during psoriasis, leading to the activation of Cdc42, NF-κB signaling pathway, and NLRP3 inflammatory bodies. These findings indicate that the myeloid cell PGGT1B plays an important protective role in the pathogenesis of psoriasis, and its absence leads to an out-of-control inflammatory response, thus aggravating the condition of psoriasis. This study not only provides a new perspective and thinking for the study of the pathogenesis of psoriasis but also provides a theoretical basis for developing new treatment strategies.

## Figures and Tables

**Figure 1 ijms-26-04901-f001:**
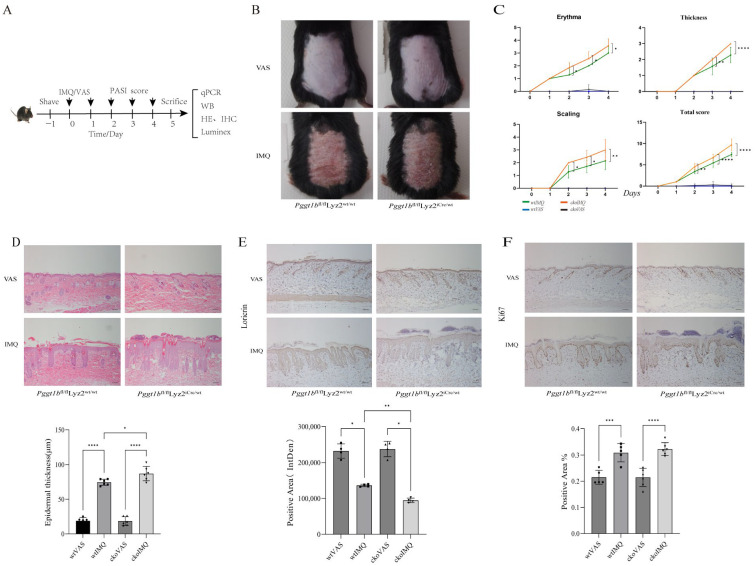
Myeloid Pggt1b deficiency aggravated IMQ-induced psoriatic rash in mice. (**A**) Flow chart of the experiment. (**B**) All wt and myeloid Pggt1b deficiency mice were treated with either Vaseline or IMQ cream. Representative images of dorsal skin lesions are presented. (**C**) PASI score progression of the above four groups from day 0 to day 5. (**D**) Representative H&E-stained images of skin sections. The results of statistical analysis of the epidermal thickness are shown. (**E**) An immunohistochemistry study showed the marker of keratinocyte differentiation (Loricrin). (**F**) An immunohistochemistry study showed the marker of keratinocyte proliferation (Ki-67). The results of the statistical analysis of the positive area percentage of relevant proteins are shown. * *p* < 0.05, ** *p* < 0.01, *** *p* < 0.001, **** *p* < 0.0001. Scale bar = 100 μm. n = 3; the data represent the average SD of three independent experiments, and it passes Shapiro–Wilk normality test and Levin variance homogeneity test. Abbreviations: wt: wild type; cko: conditional knockout; VAS: Vaseline; IMQ: imiquimod.

**Figure 2 ijms-26-04901-f002:**
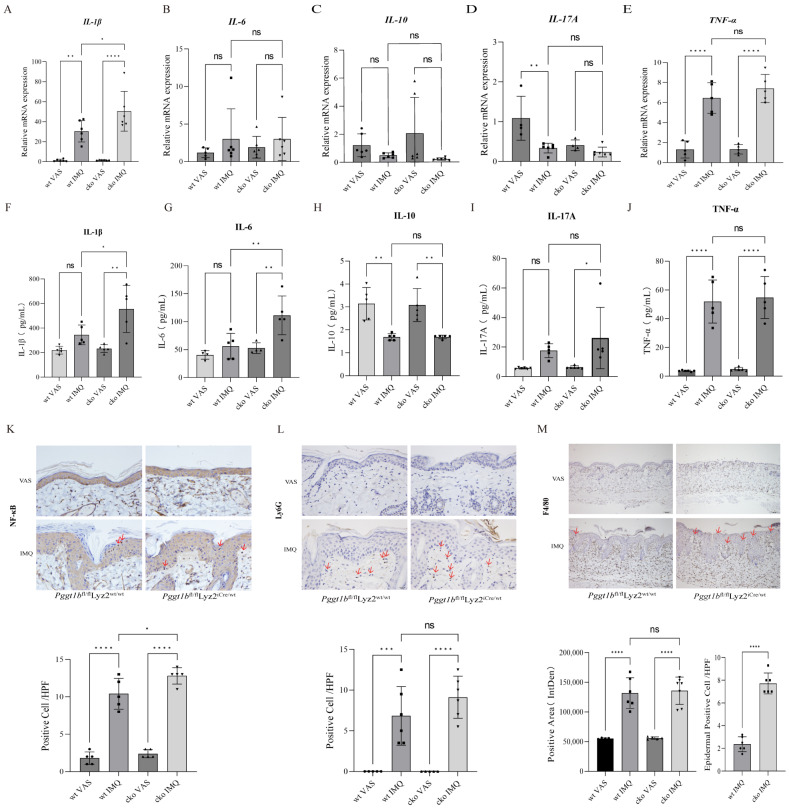
Myeloid PGGT1B deficiency exacerbated IMQ-induced psoriasiform inflammation. (**A**–**E**) qRT-PCR analysis was performed to determine the mRNA levels of IL-1β, IL-6, IL-10, IL-17A, and TNF-α. The results of the statistical analysis are shown. (**F**–**J**) Luminex was used to determine the expression levels of cytokines IL-1β, IL-6, IL-10, IL-17A, and TNF-α in the skin lesions. The results of the statistical analysis are shown. (**K**) Immunohistochemistry staining of normal and IMQ-treated skin using NF-kB (p65). The results of the statistical analysis of NF-kB translocation to the nucleus, measured as cells per high-power field, are shown. Scale bar = 20 μm. (**L**) Neutrophil infiltration was evaluated by detecting Ly6G through an immunohistochemistry study. Statistical analysis of positive cells per high-power field is shown. Scale bar = 20 μm. (**M**) Macrophage infiltration was evaluated by detecting F4/80 through an immunohistochemistry study. Statistical analysis of mean staining intensity is shown. * *p* < 0.05, ** *p* < 0.01, *** *p* < 0.001, **** *p* < 0.0001. Scale bar = 100 μm. n = 3; the data represent the average SD of three independent experiments. They pass Shapiro–Wilk normality test and Levin variance homogeneity test. Abbreviations: wt: wild type; cko: conditional knockout; VAS: Vaseline; IMQ: imiquimod.

**Figure 3 ijms-26-04901-f003:**
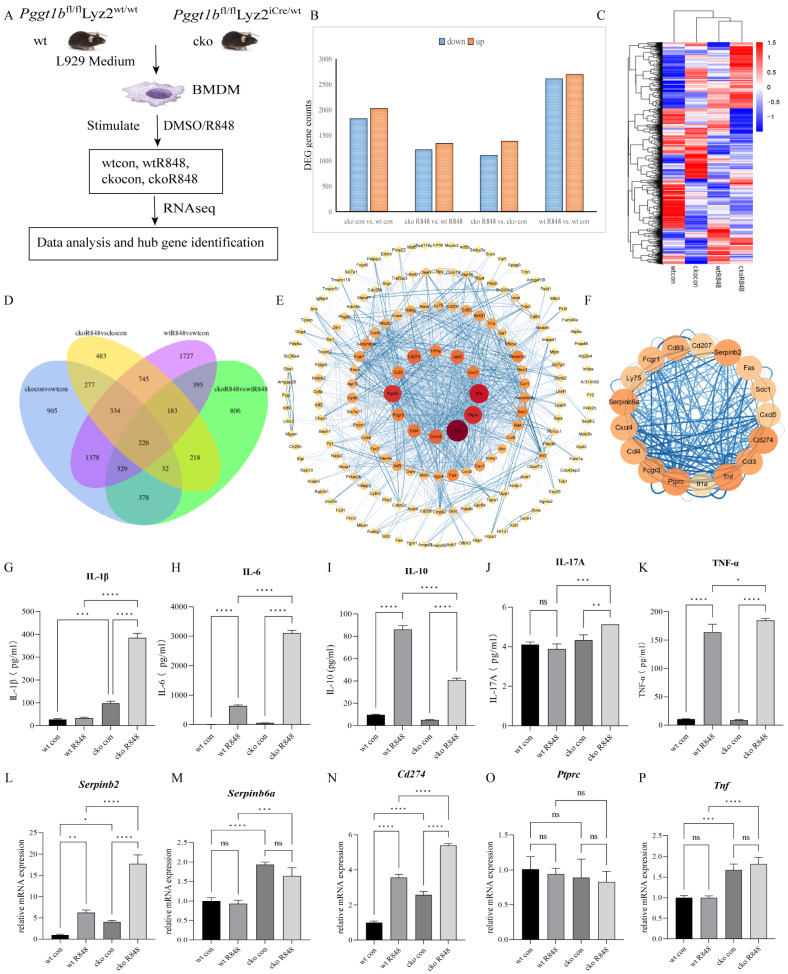
Pggt1b deficiency can promote the secretion of proinflammatory factors and inhibit the secretion of anti-inflammatory factors. (**A**) Schematic workflow of the experiment. (**B**) Statistical bar graph of differentially expressed genes. Blue and orange represent downregulated and upregulated differential genes, respectively. (**C**) Differentially expressed gene clustering heat map. The horizontal coordinate is the group, and the vertical coordinate is the normalized FPKM. The FPKM values are normalized per row, and the color ranges from dark blue (low expression) to dark red (high expression). (**D**) Venn diagram of the 4 groups of differentially expressed genes. Different colors represent different comparison combinations. (**E**) Protein interaction network diagram. The size of the nodes indicates the degree, which denotes the average number of interactions (at the score threshold) that a protein has in the network. The thickness of the edge indicates the combined score, which represents the confidence of the link between two proteins. (**F**) MCODE cluster1. Score = 12.375. (**G**–**K**) Luminex was used to determine the levels of cytokines Il1b, Il6, Il10, Il17a, and Tnfa in the BMDM culture supernatants. The results of the statistical analysis are shown. (**L**–**P**) Quantitative real-time PCR analysis was performed to determine the mRNA levels of the top 5 MCODE cluster1 genes. The results of the statistical analysis are shown. * *p* < 0.05, ** *p* < 0.01, *** *p* < 0.001, **** *p* < 0.0001, ns, no significant difference. n = 3; the data represents the average SD of three independent experiments. They passed Shapiro–Wilk normality test and Levin variance homogeneity test. Abbreviations: wt: wild type; cko: conditional knockout; VAS: Vaseline; R848: resiquimod.

**Figure 4 ijms-26-04901-f004:**
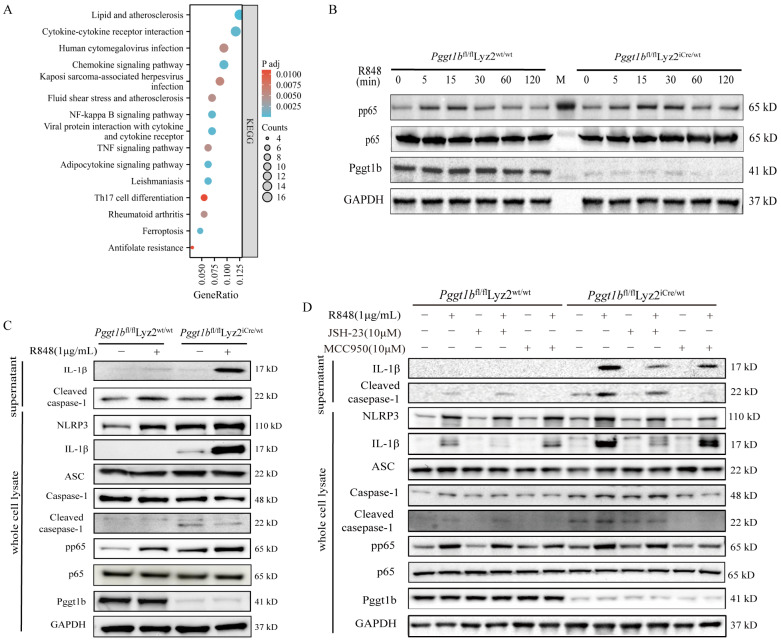
Activation of the NF-κB pathway promoted the expression of inflammasomes NLRP3 and IL-1β. (**A**) KEGG bubble chart of DEGs. (**B**) Proteins pp65, p65, and Pggt1b in cell lysates were detected using Western blotting assays, and GAPDH served as a loading control for cell plasma protein. (**C**) Proteins NLRP3, IL-1β, ASC, caspase-1, cleaved-caspase-1, and pp65 in the lysates of cell plasma and cleaved-IL-1β and cleaved-caspase-1 in the supernatants were detected using Western blotting assays. (**D**) Proteins NLRP3, IL-1β, ASC, caspase-1, cleaved-caspase-1, and pp65 in the lysates of cell plasma and cleaved-IL-1β and cleaved-caspase-1 in the supernatants with either JSH-23 (10 μM) or MCC950 (10 μM) were detected using Western blotting assays. n = 3; the data represent the average SD of three independent experiments, and they passed Shapiro–Wilk normality test and Levin variance homogeneity test. Abbreviations: wt: wild type; R848: resiquimod.

**Figure 5 ijms-26-04901-f005:**
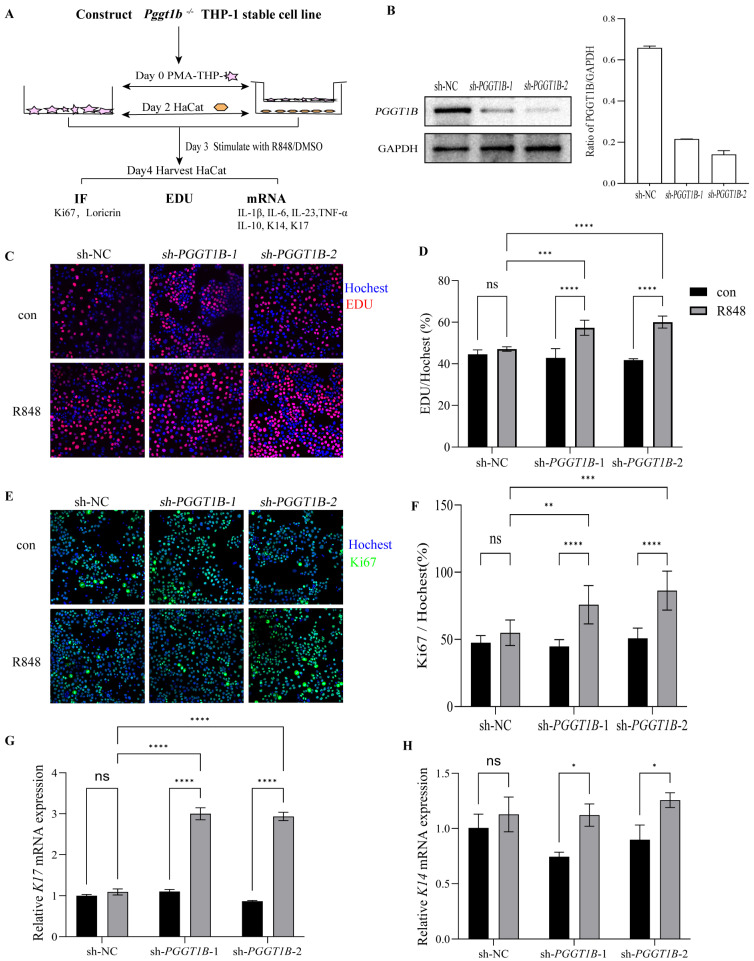
PGGT1B-deficient macrophages promoted the proliferation of HaCaT in the contact culture system. (**A**) Flow chart of the experiment. (**B**) Western blot (WB) was performed to detect PGGT1B and GAPDH as a loading control to show the knockdown of PGGT1B in THP1. (**C**) Representative EDU (red) immunofluorescent images, 200×. (**D**) Statistical analysis results of the EdU-positive rate are shown. (**E**) Immunofluorescence analysis of proliferation marker Ki67 (green), 200×. (**F**) Statistical analysis results of the Ki67-positive rate are shown. (**G**,**H**) Quantitative real-time PCR analysis was performed to determine the mRNA levels of K17 and K14. Statistical analysis results are shown. * *p* < 0.05, ** *p* < 0.01, *** *p* < 0.001, **** *p* < 0.0001. n = 3; the data represent the average SD of three independent experiments, and they passed Shapiro–Wilk normality test and Levin variance homogeneity test. Abbreviations: R848: resiquimod.

**Figure 6 ijms-26-04901-f006:**
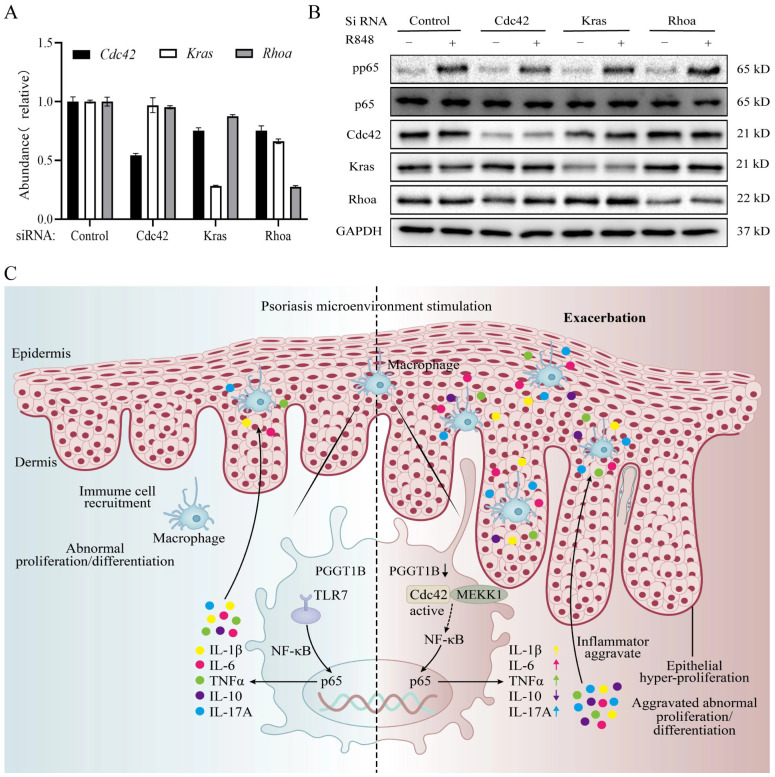
Cdc42 activation mediated increased secretion of proinflammatory factors in PGGT1B-deficient BMDMs. Mechanism diagram of macrophage PGGT1B aggravating psoriasis. (**A**) The interference efficiency of three specific siRNAs based on qRT-PCR in PGGT1B-deficient BMDMs. (**B**) Proteins pp65, p65, Cdc42, Kras, and Rhoa in the lysates of cell plasma in PGGT1B-deficient BMDMs with R848 or without were detected using Western blotting assays. (**C**) In the psoriasis microenvironment, Pggt1b-deficient macrophages are more likely to migrate into the epidermis, and Cdc42 is more likely to be activated with over-activation of the NF-κB pathway. This promotes the activation of the NLRP3 inflammatory vesicles and secretion of proinflammatory cytokines IL-1β and IL-6, which further act on keratinocytes to promote their proliferation and inhibit their differentiation, exacerbating psoriasis. n = 3; the data represent the average SD of three independent experiments, and they passed Shapiro–Wilk normality test and Levin variance homogeneity test. Abbreviations: R848: resiquimod.

**Table 1 ijms-26-04901-t001:** Hub genes in cluster 1.

Gene ID	Gene Name	Gene Description	MCODE Score
ENSMUSG00000000982	Ccl3	chemokine (C-X-C motif) ligand 3	9.34
ENSMUSG00000018930	Ccl4	chemokine (C-C motif) ligand 4	9.34
ENSMUSG00000045382	Cxcr4	chemokine (C-X-C motif) receptor 4	9.01
ENSMUSG00000016496	Cd274	CD274 antigen	8.76
ENSMUSG00000059498	Fcgr3	Fc receptor, IgG, low affinity III	8.76
ENSMUSG00000029371	Cxcl5	chemokine (C-X-C motif) ligand 5	8.51
ENSMUSG00000015947	Fcgr1	Fc receptor, IgG, high affinity I	8.19
ENSMUSG00000015396	Cd83	CD83 antigen	8.01
ENSMUSG00000034783	Cd207	CD207 antigen	8.0
ENSMUSG00000060147	Serpinb6a	serine (or cysteine) peptidase inhibitor, clade B, member 6a	7.91
ENSMUSG00000062345	Serpinb2	serine (or cysteine) peptidase inhibitor, clade B, member 2	7.91
ENSMUSG00000024401	Tnf	tumor necrosis factor	7.78
ENSMUSG00000026395	Ptprc	protein tyrosine phosphatase, receptor type, C	7.78
ENSMUSG00000027399	Il1a	interleukin 1 alpha	7.64
ENSMUSG00000024778	Fas	Fas (TNF receptor superfamily member 6)	7.56
ENSMUSG00000026980	Ly75	lymphocyte antigen 75	7.56
ENSMUSG00000020592	Sdc1	syndecan 1	7.56

**Table 2 ijms-26-04901-t002:** Primer sequence.

Gene	Sequences	Product Size (bp)
IL-1β	F: TGCCACCTTTTGACAGTGATG R: TGATGTGCTGCTGCGAGATT	138
IL-6	F:GACAAAGCCAGAGTCCTTCAGA R:TGTGACTCCAGCTTATCTCTTGG	76
IL-10	F:CCAAGGTGTCTACAAGGCCA R;GCTCTGTCTAGGTCCTGGAGT	136
IL-17A	F:TCTTTAACTCCCTTGGCGCA R:TCAGGGTCTTCATTGCGGTG	100
TNF-α	F:GATCGGTCCCCAAAGGGATG R:CCACTTGGTGGTTTGTGAGTG	92
K17	F:CACCATCCGCCAGTTTACCT R:AGGTCCGAGATGAACCTCCA	74
K14	F:GAACCACGAGGAGGTGGC R:TTCATGCTGAGCTGGGACTG	120
Kras	F:AGCAGTCAACAAAACAAGTCAGA R:AACTCTTCTCTTTTCCCCCAAT	73
RhoA	F:GTCGGGAGTTGGACTAGGCA R:AACCCTCACTGTCTTCACCC	107
Cdc42	F:GGCGGAGAAGCTGAGGAC R:ACCAACAGCACCATCACCAA	110
GAPDH	F:TGCAACCGGGAAGGAAATGA R:GCATCACCCGGAGGAGAAAT	148

## Data Availability

The datasets used and/or analyzed during the current study are available from the corresponding author upon reasonable request.

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
