# Peer review of "Myeloid PGGT1B Deficiency Promotes Psoriasiform Dermatitis by Promoting the Secretion of Inflammatory Factors"

_ijms, 2025, doi:10.3390/ijms26104901_

Round 1
Reviewer 1 Report
Comments and Suggestions for Authors
The manuscript presents an original view about the importance of PGGT1B in the pathogenesis of psoriasis. The research design and the experimental data are adequate and well acquired, the analysis of the results is well performed, and the conclusions are appropriate in relation to the results. However, the manuscript is not appropriate for publication in IJMS, requiring a major revision, rewriting, and attention to the following aspects.
1 . The manuscript has major organization problems in the Introduction, Discussion and Conclusion. These need improvement.
2. In the Introduction, it is not clear what the hypothesis/aim of the study is, and the experimental approach and main results should be included.
3. In the Discussion, the authors should start by answering the question asked in the Introduction, i.e. what are the main findings of the study. The first two paragraphs about psoriasis and PGGT1B are a literature review that should never be included in a Discussion. I suggest removing these paragraphs from the Discussion and organizing them with the Introduction paragraphs. The Discussion should also include the strengths and limitations of the study and the ideas for future research. The authors make a confusion in mentioning these topics in the Conclusion.
3. In the Conclusion, the authors wrongly mention the limitations of the study. These should only be mentioned in the Discussion. The authors should provide a clear take-home message of the study.
5. The Materials and Methods section should be reviewed in terms of language, spelling, word use, and grammar, in particular in the Supplementary Information document, where it is not well-written according to scientific writing rules.
4. Several acronyms are not well presented, spelled, and defined throughout the study. For example, PGGT1B is often written as PGTT1B in the abstract and the main text. I suggest reviewing the entire manuscript to correct these errors.
Comments on the Quality of English LanguageYour manuscript requires revision with respect to the language used. I therefore suggest that you ask a native English speaker or equivalent to assist you with correcting the spelling, grammar, word use, and punctuation throughout your manuscript.
Author Response
The manuscript presents an original view about the importance of PGGT1B in the pathogenesis of psoriasis, The research design and the experimental data are adequate and well acquired, the analysis of the results is well performed, and the conclusions are appropriate in relation to the results, However, the manuscript is not appropriate for publication in lJMS, requiring a major revision, rewriting, and attention to the following aspects.
- The manuscript has major organization problems in the Introduction, Discussion and These need improvement.
Reply: Thank you for your comments. We have improved the structure of introduction, discussion and conclusion.
- In the Introduction, it is not clear what the hypothesis/aim of the study is, and the experimentapproach and main results should be included.
Reply: Thank you for your comments. We have added the content related to the research purpose.
- In the Discussion, the authors should start by answering the question asked in theIntroduction, i.e. what are the main findings of the study. The first two paragraphs about psoriasis and PGGT1B are a literature review that should never be included in a Discussion. l suggest removing these paragraphs from the Discussion and organizing them with the introduction paragraphs, The Discussion should also include the strengths and limitations of the study and the ideas for future research, The authors make a confusion in mentioning these topics in the
Reply: Thank you for your comments. We have revised the discussion section according to the comments.
- In the Conclusion, the authors wrongly mention the limitations of the study. These should onlybe mentioned in the Discussion. The authors should provide a clear take-home message of the
Reply: Thank you for your comments. We have deleted the related content of research limitations in the conclusion part.
- The Materials and Methods section should be reviewed in terms of language, spelling, worduse, and grammar, in particular in the Supplementary Information document, where it is not well-.written according to scientific writing rules.
Reply: Thank you for your comments. We have reviewed the language, spelling, words and grammar of the article.
- Several acronyms are not well presented, spelled, and defined throughout the study. Forexample, PGGT1B is often written as PGTT1B in the abstract and the main text. l suggest reviewing the entire manuscript to correct these errors.
Reply: Thank you for your comments. We have corrected the mistakes in abbreviations.
Comments on the Quality of English Language
Your manuscript requires revision with respect to the language used. l therefore suggest that you ask a native English speaker or equivalent to assist you with correcting the spelling, grammar,word use, and punctuation throughout your manuscript.
Reply: Thank you for your comments. We have asked people with the same level of English to correct spelling, grammar, words and punctuation in the manuscript.
Reviewer 2 Report
Comments and Suggestions for Authors
This manuscript presents a comprehensive and well-structured investigation into the role of PGGT1B in psoriasis pathogenesis. The authors utilized a conditional knockout mouse model, bone marrow-derived macrophages (BMDMs), in vitro co-culture systems, transcriptomic profiling, and multiple molecular biology assays to demonstrate that PGGT1B deficiency in myeloid cells promotes NF-κB pathway activation, enhances NLRP3 inflammasome activity, and leads to increased keratinocyte proliferation. The work is novel, methodologically sound, and addresses an important question in inflammatory skin disease research. However, several aspects of the manuscript require revision before it can be considered for publication.
Major:
- Terminology Consistency and Nomenclature: Throughout the manuscript, there are inconsistent usages and typographical errors in gene naming. For instance, “PGTT1B” is mistakenly used instead of “PGGT1B” in several locations. Please standardize gene and protein nomenclature (e.g., PGGT1B, CKO vs. cko, WT vs. wt) throughout the text, figures, and legends, according to current HGNC/Mouse Genome Informatics guidelines.
- Figure Legends and Statistical Reporting: The figure legends lack sufficient detail. Please specify sample size (n), statistical tests used, and whether error bars represent standard deviation (SD) or standard error of the mean (SEM). Indicate whether data passed assumptions for parametric testing (normality, variance homogeneity), especially for ANOVA.
- Language and Style: The manuscript would benefit significantly from professional English editing to improve readability, remove redundancies (e.g., "Psoriasis, commonly known as psoriasis..."), and enhance scientific clarity.
- Clinical Relevance and Human Data: While the in vivo and in vitro data are convincing, the authors should strengthen the translational relevance by discussing or including expression profiles of PGGT1B in human psoriatic skin or PBMC datasets (e.g., GEO, ArrayExpress, or other published RNA-seq datasets). If human validation is not feasible at this stage, please clearly acknowledge this limitation and propose it as a future direction.
- Mechanistic Depth and Clarifications: The role of CDC42 is intriguing. However, more discussion is needed on how PGGT1B-mediated geranylgeranylation regulates CDC42 activation. Is this direct or indirect? Has protein prenylation of CDC42 been confirmed? Similarly, clarify the rationale for selecting R848 as the TLR agonist for BMDM stimulation, rather than IMQ, which was used in vivo.
Minor Suggestions:
- Add gene accession numbers where relevant (e.g., in RNA-seq data).
- Improve the flow of the Abstract to avoid repetitive phrasing and ensure clear objectives and outcomes.
- Confirm the availability of raw data (e.g., RNA-seq FASTQ files) in a public repository in compliance with IJMS data policies.
Author Response
Comments and Suggestions for Authors
This manuscript presents a comprehensive and well-structured investigation into the role ofPGGT1B in psoriasis pathogenesis. The authors utilized a conditional knockout mouse model,bone marrow-derived macrophages (BMDMs), in vitro co-culture systems, transcriptomic profiling, and multiple molecular biology assays to demonstrate that PGGT1B deficiency inmyeloid cells promotes NF-kB pathway activation, enhances NLRP3 inflammasome activity, andleads to increased keratinocyte proliferation. The work is novel, methodologically sound, and addresses an important question in inflammatory skin disease research. However, several aspects of the manuscript require revision before it can be considered for publication.
Major
Terminology Consistency and Nomenclature: Throughout the manuscript, there areinconsistent usages and typographical errors in gene naming. For instance, "PGTT1B” ismistakeniy used instead of"PGGT1B" in several locations. Please standardize gene and protein nomenclature (e.g., PGGT1B, CKO vs. cko, WT vs. wt) throughout the text. Fiqures and legends, according to current HGNC/Mouse Genome Informatics guidelines.
Reply: Thank you for your comments. We have revised this question.
Figure Legends and Statistical Reporting: The figure legends lack sufficient detail. Please specify sample size (n), statistical tests used, and whether error bars represent standard deviation (SD) or standard error of the mean (SEM). Indicate whether data passed assumptions for parametric testing (normality, variance homogeneity), especially for ANOVA.
Reply: Thank you for your comments. We have added details about the legend.
Language and Style: The manuscript would benefit significantly from professional English editing to improve readability, remove redundancies (e.g., "Psoriasis, commonly known as psoriasis..."), and enhance scientific clarity.
Reply: Thank you for your comments. We have revised the language of the article.
Clinical Relevance and Human Data: While the in vivo and in vitro data are convincing, the authors should strengthen the translational relevance by discussing or including expression profiles of PGGT1B in human psoriatic skin or PBMC datasets (e.g., GEO, Array Express, or other published RNA-seq datasets). if human validation is not feasible at this stage, please clearly acknowledge this limitation and propose it as a future direction.
Reply: Thank you for your comments. We have added this limitation in the discussion section.
Mechanistic Depth and Clarifications: The role of CDC42 is intriguing. However, morediscussion is needed on how PGGT1B-mediated geranylgeranylation regulates CDC42activation, ls this direct or indirect? Has protein prenylation of CDC42 been confirmed? Similarly, clarify the rationale for selecting R848 as the TLR agonist for BMDM stimulation,rather than lMQ, which was used in vivo.
Reply: Thank you for your comments. We have clarified the above contents in the discussion section.
Minor Suggestions:
Add gene accession numbers where relevant (e.g. in RNA-seg data).
Reply: Thank you for your comments. Table 1 shows the gene accession number.
Improve the flow of the Abstract to avoid repetitive phrasing and ensure clear objectives and outcomes.
Reply: Thank you for your comments. We have revised the summary.
Confirm the availability of raw data (e.g., RNA-seq FASTQ files) in a public repository in compliance with lJMS data policies.
Reply: Thank you for your comments. We have confirmed the relevant information.
Round 2
Reviewer 1 Report
Comments and Suggestions for Authors
The authors have addressed all the comments. I recommend publication of the manuscript after minor revision of the Supplementary Information. The Supplementary Information requires revision with respect to the language used.
Comments on the Quality of English LanguageThe Supplementary Information requires revision with respect to the language used. l suggest that you ask a native English speaker or equivalent to assist you with correcting the spelling, grammar, word use, and punctuation throughout the Supplementary Information.
Author Response
Comments and Suggestions for Authors
The authors have addressed all the comments. recommend publication of the manuscriot afterminor revision of the Supplementary Information. The Supplementary Information requiresrevision with respect to the language used.
Reply: Thank you for your comment. We have improved the language of supplementary materials.
The Supplementary Information requires revision with respect to the language used. l suggesthat you ask a native English speaker or equivalent to assist you with correcting the spelling,grammar, word use, and punctuation throughout the Supplementary Information.
Reply: Thank you for your comment. We have improved the language of supplementary materials.
Reviewer 2 Report
Comments and Suggestions for Authors
I appreciate the authors' efforts in addressing the comments raised in the first round of review. The revised manuscript demonstrates significant improvement in clarity, structure, and scientific rigor. Most of the critical concerns—including nomenclature consistency, statistical transparency, mechanistic clarification, and the discussion of translational relevance—have been appropriately addressed. However, a few minor issues remain that should be resolved prior to final acceptance:
- Abstract language refinement: The opening sentence remains slightly generic. Consider refining for greater scientific precision.
- Provide at least one supporting reference indicating whether CDC42 is known to be geranylgeranylated. Or, clearly state that direct prenylation of CDC42 has not yet been experimentally confirmed, and frame this as a limitation/future direction.
- Although figure legends have been significantly improved, it would be helpful to explicitly define key abbreviations (e.g., wt, cko, VAS, IMQ) at the beginning of each legend or as part of a footnote, for clarity to non-specialist readers.
Author Response
Comments and Suggestions for Authors
I appreciate the authors' efforts in addressing the comments raised in the first round of review. The revised manuscript demonstrates significant improvement in clarity, structure, and scientific rigor. Most of the critical concerns—including nomenclature consistency, statistical transparency, mechanistic clarification, and the discussion of translational relevance—have been appropriately addressed. However, a few minor issues remain that should be resolved prior to final acceptance.
- Abstract language refinement: The opening sentence remains slightly generic. Considerrefining for greater scientific precision.
Reply: Thank you for your comment. We have revised the beginning of the abstract.
- Provide at least one supporting reference indicating whether CDC42 is known to be Or, clearly state that direct prenylation of CDC42 has not yet been experimentally confirmed, and frame this as a limitation/future direction.
Reply: Thank you for your comment. At the end of the discussion, we have explained that the direct prenylation of CDC42 has not been confirmed by experiments, and regard it as the limitation and future direction.
- Although figure legends have been significantly improved, it would be helpful to explicitlydefine key abbreviations (e.g., wt, cko, VAS, lMQ) at the beginning of each legend or as partof a footnote, for clarity to non-specialist readers.
Reply: Thank you for your comment. We have added footnotes for key abbreviations to the legend.